# The Role of Concomitant Nrf2 Targeting and Stem Cell Therapy in Cerebrovascular Disease

**DOI:** 10.3390/antiox11081447

**Published:** 2022-07-26

**Authors:** Jonah Gordon, Gavin Lockard, Molly Monsour, Adam Alayli, Cesario V. Borlongan

**Affiliations:** Center of Excellence for Aging and Brain Repair, Department of Neurosurgery and Brain Repair, University of South Florida Morsani College of Medicine, Tampa, FL 33602, USA; jonahgordon@usf.edu (J.G.); gavinlockard@usf.edu (G.L.); mmonsour@usf.edu (M.M.); adamalayli@usf.edu (A.A.)

**Keywords:** Nrf2, stem cell, stroke, cerebrovascular disease, peripheral inflammation

## Abstract

Despite the reality that a death from cerebrovascular accident occurs every 3.5 min in the United States, there are few therapeutic options which are typically limited to a narrow window of opportunity in time for damage mitigation and recovery. Novel therapies have targeted pathological processes secondary to the initial insult, such as oxidative damage and peripheral inflammation. One of the greatest challenges to therapy is the frequently permanent damage within the CNS, attributed to a lack of sufficient neurogenesis. Thus, recent use of cell-based therapies for stroke have shown promising results. Unfortunately, stroke-induced inflammatory and oxidative damage limit the therapeutic potential of these stem cells. Nuclear factor erythroid 2-related factor 2 (Nrf2) has been implicated in endogenous antioxidant and anti-inflammatory activity, thus presenting an attractive target for novel therapeutics to enhance stem cell therapy and promote neurogenesis. This review assesses the current literature on the concomitant use of stem cell therapy and Nrf2 targeting via pharmaceutical and natural agents, highlighting the need to elucidate both upstream and downstream pathways in optimizing Nrf2 treatments in the setting of cerebrovascular disease.

## 1. Introduction

There are two major types of cerebrovascular accidents, hemorrhagic (HS) and ischemic (IS) stroke. IS refers to an obstruction of blood flow to a region in the brain, ultimately leading to significant CNS damage. HS involves blood vessel damage and rupture in the cerebral vasculature, also resulting in substantial CNS damage. Approximately 87% of strokes are ischemic, with the remainder being hemorrhagic [1]. Nearly a quarter of patients with IS die, and half of patients with HS [2]. One-sixth of people around the world will have a stroke in their lifetime, and the yearly incidence of stroke is nearly 14 million. Furthermore, it is the fifth leading cause of death in the United Sates and the leading cause of long-term disability [3,4,5]. Despite these grave statistics, treatments for both types of strokes are severely limited. For IS, there are narrow time windows where thrombolytic therapies, such as tissue plasminogen activator (tPA) or mechanical thrombectomy (MT) can be employed. If these treatments are attempted outside of a 4.5 h time window, hemorrhagic transformation can occur and further detriment a patient’s prognosis [6,7,8,9,10]. HS has no proven clinical therapies [11]. With poor prognoses and little to no treatment available, it is vital that greater research focus is attributed to stroke therapy development.

While substantial brain damage occurs because of IS or HS conditions alone, it can be argued that an even greater contributor to stroke pathophysiology is secondary cell death mechanisms [3,12]. After both types of strokes, major cell death mechanisms include excitotoxicity, oxidative stress, free radical accumulation, mitochondrial dysfunction, inflammation, and impaired neurogenesis, angiogenesis, and vasculogenesis [13,14,15]. With such a wide range of pathophysiological mediators, ample treatment targets exist to be studied as potential stroke therapeutics. Nuclear factor erythroid 2-related factor 2 (NFE2L2/Nrf2) and its associated pathway is one such attractive target due to its role in sequestration of oxidative stress that will remain as the focus of this review.

Nrf2 binds to promoters of antioxidant genes and thus serves a crucial role in the attenuation of reactive oxygen/nitrogen species. Seeing that the brain is vulnerable to oxidative stress [16], activation of Nrf2 is an attractive target in the setting of ischemia such as cerebrovascular accidents. An E3 ubiquitin ligase complex consisting of several proteins including Kelch-like ECH-Associated Protein 1 (KEAP1), Cullin 3 (CUL3), and RING-box protein 1 (RBX1) serves to inhibit Nrf2 in the normal physiological setting. At baseline, the complex binds to Nrf2 for ubiquitination and subsequent proteasomal degradation of Nrf2 [17]. In the event of oxidative stress, KEAP1′s conformation adjusts via modification of cysteine thiols [18], allowing Nrf2 to accumulate. Nrf2 then travels to the cell nucleus, binds with small musculoaponeurotic fibrosarcoma (sMaf) proteins, and drives transcription of >250 cytoprotective genes. Many genes driven by Nrf2 are involved with glutathione synthesis or action. De novo synthesis of glutathione begins with the action of the enzyme γ-glutamylcysteine synthetase, which uses ATP hydrolysis to combine glutamate with cysteine to produce γ-glutamylcysteine. The enzyme glutathione synthetase then utilizes ATP to add glycine to the dipeptide to form glutathione [19,20]. Glutathione conjugates to xenobiotics to eliminate them; glutathione S-transferases (GSTs) are enzymes that assist in this manner [21]. Glutathione must be in its reduced form to act properly, which the enzyme glutathione reductase ensures [22]. Other examples of Nrf2 promoted genes include superoxide dismutase (SOD), catalase, heme oxygenase 1 (HO-1), and NAD(P)H quinone dehydrogenase 1 (NQO1).

Nrf2 plays a vital role in managing excessive oxidative stress following stroke, with a vital role in heme and iron metabolism, antioxidant proliferation, glutathione regeneration, thioredoxin, and protein recycling [23,24]. The major producers of Nrf2 in the brain include microglia and macrophages [25,26]. In fact, microglia and astrocytes produce an extreme amount of Nrf2 within 24 h following an induced middle cerebral artery occlusion (MCAO) [27]. In IS, hydrogen peroxide is elevated after ischemia and reperfusion, serving as a major stimulus for Nrf2 activation [5,28,29,30,31]. IS models in Nrf2-/- rats have greater damage after IS due to the loss of the protective Nrf2 properties [28,32,33]. While targeting Nrf2 has clear benefits for IS treatment, similar toxic ROS accumulation in HS suggests that Nrf2 may also be beneficial for HS treatment. HS Nrf2-/- models have poor prognoses, with larger hematomas, more functional defects, greater cell death, ROS, and cell damage [34,35]. The primary contributors heme, hemoglobin, and iron have been shown to induce Nrf2 signaling, further elucidating this signaling mechanism’s role in lessening oxidative damage following HS [36,37]. Within its producer cells, Nrf2 downstream proteins clear red blood cell debris and reduce oxidative damage [38]. One Nrf2-related protein mentioned above, HO-1, is an antioxidant which produces protective substances, carbon monoxide and biliverdin, and increases microglial capabilities to phagocytose HS damaged tissue [36,39]. HO-1, the stress-induced isoform of heme oxygenase, is significantly increased within 3–5 days following HS in rodents [40,41]. Other than HO-1 simply degrading proinflammatory heme into anti-inflammatory carbon monoxide and eventual bilirubin, it also decreases NF-κB signaling [42], which subsequently decreases proinflammatory cytokines including TNF-α and IL-6. Targeting any of the aforementioned ameliorators of oxidative stress has therapeutic potential, however, we argue that targeting the most upstream protein, Nrf2, will show the greatest therapeutic benefit.

A schematic representation of the Nrf2 pathway, downstream effectors, and molecular crosstalk with the NF-κB pathway can be seen in Figure 1.

Despite the described therapeutic effects of Nrf2 upregulation, it must be noted that such benefits of Nrf2 are dependent on disease state. Recent reviews have pointed to uncontrolled Nrf2 expression as the reason for chemoresistance through inhibition of oxidative stress in ovarian cancer [43] and gastric cancer [44]. Thus, the therapeutic benefits of Nrf2 are limited.

This review will probe scientific evidence for the reduction in oxidative stress following stroke by targeting the Nrf2 signaling pathway.

## 2. Neuroinflammation and Oxidative Stress in Stroke

Oxidative stress is a common feature of HS and IS; however, the processes vary due to different mechanisms of infarction. In IS, a cascade of events prompted by nutrient and oxygen deprivation promote cell death. Without ample oxygen, mitochondrial respiration is shut down, and ATP levels are depleted. Thus, ATPase pumps vital to cell equilibrium, including the Na^+^/K^+^ pump, are inoperative. Further along this cascade, Na^+^/Ca^2+^ pumps are disrupted, and Ca^2+^ accumulation occurs within neurons, promoting lipases, proteases, and other cell death pathways [45]. Mitochondria undergo programmed death processes and release cytochrome C, leading to apoptosis [46]. Further expression of the Fas ligand also instigates cell death pathways. This excessive cell death increases levels of reactive oxygen species (ROS) and exacerbates mitochondrial dysfunction and ATP shortages [47,48]. Furthermore, alterations in mitochondrial metabolic processes also increase ROS. Despite an already exorbitant number of pathological processes, cell death also causes excitotoxicity due to excessive glutamate release. Glutamate furthers the ongoing processes by stimulating even more Ca^2+^ influx, perpetuating a cycle of inflammation and cell death [49]. Apoptotic neurons, necrotic neurons, and their released products serve as damage-associated molecular patterns activating a cascade of inflammatory immune cells within the infarction area. Local microglia, astrocytes, and neutrophils release ROS, perpetuating oxidative stress within the brain after IS [47,50,51]. Later invasion of peripheral immune cells further worsens oxidative stress. Ultimately, the brain is faced with an overwhelmingly toxic proliferation of ROS after IS.

While cell death and release of ROS is also seen in HS, oxidative stress following HS has additional pathologic ROS contributions, primarily related to the large influx of hemoglobin and its metabolites [12]. Once a blood vessel ruptures, red blood cells flood the cerebral parenchyma and release hemoglobin after loss of membrane integrity [52]. Hemoglobin, composed of heme, is also broken down by macrophages and microglia. Primary constituents of heme include iron, biliverdin, and carbon monoxide [36,52]. Iron and heme ultimately contribute to the oxidative damage seen after HS, causing damage to the local neurovasculature and blood–brain barrier [39,53,54,55]. Iron promotes the Fenton reaction, a notorious contributor of free radicals [36,56,57]. Furthermore, iron can promote ferroptosis, an iron-dependent cell death, further enhancing oxidative damage via ROS, lipid peroxidation, mitochondrial changes, and inflammatory pathways [55,58,59]. Both IS and HS demonstrate extreme ROS damage, perpetuating a cycle of cell death and an injurious local environment. Reducing oxidative stress should be a central therapeutic goal for stroke, possibly revolutionizing stroke treatments.

## 3. Peripheral Inflammation and Nrf2

### 3.1. Mechanistic Interrogation of Peripheral Inflammation

In addition to the drastic oxidative damage observed after stroke, inflammation undoubtedly plays a major role in the clinical and functional outcomes of patients with ischemic or hemorrhagic stroke. Growing scientific evidence indicates that stroke should be treated as a multi-organ disease, rather than the traditional compartmentalization of the CNS and other affected organ systems (Figure 2) [60]. Local inflammatory processes have been clearly documented to cause damage to brain tissue [61], and a more recent body of evidence points to peripheral inflammation as also playing an important role in the pathogenesis of stroke. For example, plasma concentrations of IL-6 were closely correlated with infarct volume and stroke severity [62]. Other inflammatory markers such as C-Reactive Protein are also elevated upon the onset of stroke [63].

Organs that influence peripheral inflammation include but are not limited to the spleen [64], cervical lymph nodes [65], and bone marrow stem cells [66]. Of note, the spleen has been shown to decrease in size following stroke in rats subjected to MCAO [64,67,68]. This is largely due to the release of lymphocytes, monocytes, and neutrophils from the spleen due to upregulated catecholamines, as seen in stroke models such as MCAO [69,70,71]. These splenocytes play a significant role in inflammation, and accordingly, rats with splenectomy prior to the MCAO display an 82.3% decrease in infarct volume compared to those with an intact spleen [64]. Clinical studies also show an inverse correlation between spleen size and blood lymphocyte counts, while there is a positive correlation with neutrophil counts, further supporting evidence of the spleen’s role in peripheral inflammation [72]. Bone marrow has a relatively less understood role in peripheral inflammation following ischemic stroke, but recent studies in mice subjected to MCAO show activation of hematopoiesis via bone marrow β3 adrenoreceptors [66]. Similar to splenectomy, the removal of cervical lymph nodes decreases the extent of brain damage in the days after ischemic stroke. Additionally, the possible activation of VEGFR3 tyrosine kinase receptors on cervical lymph nodes by VEGF-C secretion from the brain stands as a potential cause for the initiation of peripheral immune response [73], tying together the multi-organ web that carries out the damaging changes we see from peripheral inflammation.

Although much of the damage occurs due to the inflammation itself, the immunodepression that follows poses a sizable challenge that cannot be ignored. Immunodepression is the body’s response to the acute inflammation that occurs post-stroke, but it can precipitate a wide range of infections, the most common of which are pneumonia and urinary tract infections [74]. This primarily occurs through the activation of the β-arrestin2-NF-κB [75] or cAMP-PKA-NF-κB pathways [76], exciting the adrenal medulla and increasing plasma catecholamine levels which ultimately decreases the amount of plasma lymphocytes [77]. Other pathways, such as the cholinergic anti-inflammatory pathway [78] and the hypothalamus–pituitary–adrenal axis [79], also exacerbate the risk of infection.

Nrf2′s role in post-stroke peripheral inflammation is still a topic of exploration. Recent research has shown Nrf2′s role to lie primarily in its complex transcriptional regulation of inflammatory pathways such as the NF-κB system [80] and subsequently, cytokines such as IL-6 and IL-1β [81,82]. Nrf2 has also been shown to regulate other cytokines such as COX-2 and iNOS in mouse models [83], although this has not yet been shown in MCAO mice. More research needs to be conducted to fully understand Nrf2′s role in post-stroke peripheral inflammation, particularly whether it impacts any of the organ systems described above, but the general wealth of knowledge on Nrf2′s role in other pathologies gives researchers a substantial incentive to investigate Nrf2 in stroke.

### 3.2. Stem Cells and Peripheral Inflammation

While stem cell therapy for stroke is an area of heavy focus, there is comparatively less research conducted on the role of stem cells in the context of peripheral inflammation. A perceived problem was that stem cells administered peripherally could not cross the blood–brain barrier, which limits their applications. However, recent research has shown that there is potential for stem cell therapy to modulate the peripheral immune response. Infusion of human umbilical cord blood cells was shown to restore spleen size and function in rats with MCAO [84,85], coincident with significant reduction in ischemic brain damage and improved behavioral performance [86]. Subsequently, multipotent adult progenitor cells have been shown to suppress groups of genes that upregulate inflammatory response from within the spleen [87]. This resulted in decreased levels of inflammatory cytokines such as IL-1β and TNF-α. These studies all show that targeting the spleen can significantly benefit stroke outcomes. More recent research has shown that bone marrow mesenchymal stem cells implanted into rat brains with MCAO preferentially migrate to the spleen [88]. Unfortunately, less is known about the applications of stem cells in the other organs involved in peripheral immunity during a stroke. As more is revealed about the intricacies of the brain’s post-stroke communication with organs in the body, researchers will be able to develop more targeted therapies to address peripheral immune response more precisely and effectively. Considering the substantial literature on Nrf2, alongside emerging literature using cell-based therapies for stroke, combining these two therapeutic modalities may be incredibly beneficial.

## 4. Current Nrf2-Directed Treatment Modalities

### 4.1. Stem Cell Therapies and Nrf2

Several initiatives have been implemented to examine the therapeutic role of exogenous stem cells for stroke treatments [13,89]. It is hypothesized that these cell-based therapies work via a bystander effect [90]. As opposed to the transplanted cells replacing the dead or dying ischemic cells, bystander effects refer to the grafted stem cells releasing protective factors against neuroinflammation and oxidative stress [91,92,93]. This ultimately promotes neurogenesis, angiogenesis, oligodendrogenesis, vasculogenesis, and synaptogenesis by releasing anti-inflammatory and repair signals [94]. Clinically and pre-clinically, stem cells such as fetal-derived neural stem cells, embryonic stem cells, bone marrow stem cells, umbilical cord stem cells, adipose stem cells, and induced pluripotent stem cells have been examined for stroke treatment and demonstrate promising results [74,95,96,97,98,99]. Unfortunately, the use of stem cells after stroke poses some difficulty due to the exorbitant inflammatory and oxidative stressors present within the infarcted area. Oxidative stress can induce autophagy, or cell-induced recycling of intracellular components. While beneficial and homeostatic in some instances, this process of ROS-induced autophagy can be of detriment to stem cell viability [100]. To optimize stem cell treatments and amplify cell survival for the most beneficial results, oxidative stress should be reduced prior to or concurrently with stem cell administration.

Endogenously, adult human neurogenesis is incredibly limited to only certain locations of the brain such as the subgranular zone of the dentate gyrus and subventricular zone of the lateral ventricles [101], but neural stem cells (NSCs) do reside there and are able to differentiate into neurons and certain glia, including oligodendrocytes and astrocytes [102,103]. NSC survival and differentiation is promoted by Nrf2 secondary to its role in the reduction in intracellular ROS [104,105]. In rats, it has been noted that variations in expression levels of Nrf2 are tied with the survival of NSCs [106], and in mice, a deficiency in Nrf2 expression or function resulting in the reduction in NSC proliferation could be improved with administration of Nrf2 [107]. However, following an ischemic cerebrovascular accident, neurogenesis by NSCs is insufficient for complete recovery due to relatively low yields of new cells [108,109]. Thus, a possible therapy is to administer exogenous NSCs obtained from neuroectoderm in fetuses or from the neurogenic sites in adults, after they have been cultured and exposed to basic fibroblast growth factor and epidermal growth factor [110,111]. Unfortunately, the host brain often rejects transplant NSCs [112], although it has been observed that delivery of human NSC-derived extracellular vesicles improves transplantation outcomes by stimulating nuclear localization of Nrf2, which subsequently decreases oxidative stress and stimulates axon elongation [113,114]. The following section investigates drug therapy that may be used concomitantly with NSC therapy to improve outcomes in patients suffering from cerebrovascular disease.

### 4.2. Targeting Nrf2 to Enhance Stem Cell Therapy

While stem cell therapy is undoubtedly a promising stroke treatment [115], its overall efficacy may benefit from adjunctive therapeutics that ameliorate inflammation, such as Nrf2 treatments (Figure 3). The following reviews a number of potential agents for co-administration with NSCs as a potential treatment modality.

Dimethyl fumarate (DMF), derived from fumaric acid, is a known treatment for psoriasis and multiple sclerosis due to its immunomodulatory effects [116]. Considerable research into its precise mechanism and interactions is ongoing, however, it is known to bind KEAP1, resulting in upregulation of antioxidant genes, including HO-1 [117], γ-glutamylcysteine synthetase, glutathione synthetase, and glutathione reductase [118], following nuclear translocation of Nrf2 [119]. A study in an MCAO model on Sprague-Dawley rats demonstrated significantly reduced infarct volumes in the group treated with DMF compared to control counterparts [120]. As for its use in stem cell therapy, DMF delivered to mouse and rat models following a treatment with hydrogen peroxide to trigger oxidative stress resulted in increased NSC survival. Additionally, DMF enhanced the survival of motor neurons and reduced both the production of ROS and the rates of apoptosis following hydrogen peroxide treatment [103].

Berberine (BBR), a natural alkaloid isolated from medicinal herbs, is a known treatment for diarrhea that has recently become a multitarget drug for neurological disorders owed in part to antioxidant and anti-inflammatory properties [121]. The interaction of BBR with the Nrf2 pathway was demonstrated through siRNA inactivation of Nrf2 signaling, which diminished antioxidant effects following BBR’s administration [122,123]. In MCAO-induced stroke mice, BBR activated peroxisome proliferator-activated receptor-δ (PPARδ), which in turn upregulated Nrf2, along with other known antioxidants, to scavenge ROS in NSCs, thereby promoting their proliferation and improving recovery [124].

Carbon monoxide (CO) has recently been shown to have a significant role as an antioxidant, stimulating the bilirubin/biliverdin redox cycling system and the pentose-phosphate pathway to produce NADPH, a reducing equivalent [125]. CO plays a protective role against iron overload, making it an attractive treatment for administration with transplanted NSCs following hemorrhagic stroke. This has been previously demonstrated in mouse models through modulation of the crosstalk between Nrf2 and NF-κB, in which CO inhibited NF-κB while inducing ROS scavenging through Nrf2 activation [126].

Sulforaphane (SFN) is an isothiocyanate found in a number of cruciferous vegetables with known antioxidant, anti-inflammatory, and anti-tumor activity [127]. Specifically, the antioxidant properties of SFN are derived from well-documented activation of the Nrf2 pathway [128,129,130] including downstream activation of GSTs, NQO-1, and HO-1 [28]. MCAO-induced Sprague-Dawley rats treated with SFN demonstrated heightened Nrf2 levels within the cerebral microvasculature after 24 h, resulting in reduction in blood–brain barrier (BBB) disruption and lesion progression [131]. Later experimentation into Sprague-Dawley rats revealed that SFN treatment at concentrations less than 10 µM stimulated NSC differentiation and proliferation [132]. Thus, SFN shows significant promise in stroke treatment with and without concurrent stem cell treatment.

Doxycycline (DOX) is a tetracycline derivative known to be well-tolerated and demonstrate broad-spectrum efficacy as an antibiotic that also exhibits antioxidant and anti-apoptotic activity [133,134]. Antioxidant activity has been determined using electron paramagnetic resonance to confirm scavenging of superoxide by DOX and has been related to modulation of the Nrf2 pathway [134], while its anti-apoptotic role in neuroprotection has been attributed to inhibition of microglial activation [135]. Additionally, DOX has been tied to the upregulation of tight junctions with subsequent inhibition of the matrix metalloproteinases (MMP) MMP-2 and MMP-9 and protein kinase C delta for protection of the BBB [136]. This neuroprotective role explains the historic success of DOX in reducing damage from ischemic stroke modeled in rabbits and MCAO-induced rats [137,138]. As an aspect of stem cell treatment, DOX has been documented to upregulate Nrf2 mRNA and protein levels in NSCs for an overall increase in cell survival in preconditioned fetal rat brains [139,140,141].

Minocycline, a semisynthetic tetracycline, has also been associated with significant neuroprotection through a number of properties including antioxidation, anti-apoptosis, anti-inflammation, and vascular protection [142,143]. A systematic review and meta-analysis by Malhotra et al. of randomized control trials of minocycline application in stroke patients indicated success in reducing damage attributed to acute ischemic stroke and a need for further research into a potential role in acute intracerebral hemorrhage [143]. In a number of experimental studies on rat models, NSCs pretreated with minocycline resulted in a noted upregulation of Nrf2 and downstream genes such as NQO1 and HO-1, which correlated with increased NSC viability and proliferation [112,144]. Further, minocycline resulted in the NSCs releasing neuroprotective factors such as brain-derived neurotrophic factor, nerve growth factor, glial cell-derived neurotrophic factor, and vascular endothelial growth factor [112]. Outcomes of neuroinflammation were mitigated by minocycline, inhibiting microglial activation and counteracting proinflammatory cytokines that normally inhibit neurogenesis, restoring the neurogenic and oligodendrogliogenic potential of NSCs [145,146,147]. Minocycline has also been implicated in the success of bone marrow-derived mesenchymal stem cell transplantation in animal stroke models in which the combination of therapies results in reduced infarct size and enhanced recovery [148,149,150].

Tert-butylhydroquinone (tBHQ), a synthetic phenol, is a common food additive known to have low toxicity and antioxidant properties. Pulse-chase assay has demonstrated a stabilizing effect of tBHQ on Nrf2 in NSCs, making it an attractive molecule for research into neuroprotection from oxidative stress [151]. tBHQ has been studied extensively in cerebral ischemia, sufficiently activating Nrf2 in rat models to reduce cortical damage and sensorimotor deficits following ischemia-reperfusion [33]. Further studies into cerebral ischemia in mice reveal that activation of Nrf2 by tBHQ also enhances angiogenesis and astrocyte activation [152]. For intracerebral hemorrhages, tBHQ treatment reduced oxidative brain damage, microglial activation, and release of proinflammatory cytokines to improve outcomes in CD1 mice [153]. Models of subarachnoid hemorrhage treated with tBHQ highlighted the upregulation of KEAP1, Nrf2, HO-1 and NQO1to reduce early brain injury and cognitive dysfunction [154]. tBHQ has recently been implicated in the generation of multipotent stem cells from normal brain pericytes placed under oxidative stress for a key role in regeneration of cerebral vasculature, suggesting the potential for future research into tBHQ and in vivo generation of stem cells [155].

Resveratrol, a natural stilbene, is produced by a number of fruits and vegetables with known antioxidant and antitumor activity [156]. In response to vascular injury, resveratrol has been indicated to mobilize endothelial cells and increase endothelial progenitor cell proliferation [157,158]. A recent meta-analysis in rodents has demonstrated an overall neuroprotective effect of resveratrol in ischemic stroke models [159]. On a molecular level, this effect is achieved through upregulation of Nrf2 and downstream antioxidant genes, including HO-1 [160], that enhance neurogenesis and increase viability of NSCs by reducing apoptosis [140,161].

Ginseng, along with its bioactive ingredients ginsenosides, is a staple of herbal medicine predominantly utilized in Asia. Rodent and human models have highlighted a number of important pharmaceutical properties that include antioxidant, anti-inflammatory, and immunomodulatory effects [162]. These effects are achieved through inhibition of the proinflammatory NF-κB pathway and subsequent upregulation of Nrf2 [163]. *Panax notoginseng* was also shown to reduce the expression of Nogo-A and Nogo Receptor, molecules that typically inhibit axonal regeneration after ischemic injury [162]. As it pertains to stem cell therapy, ginseng has been implicated in regeneration during inflammatory diseases such as stroke, inducing neurogenesis and angiogenesis that has been shown to preserve cognitive function [164]. For instance, pretreatment of MCAO-induced Wistar rats with Ginseng total saponins resulted in improved neurological scores after a two-week recovery period, highlighting Ginseng’s inductive effect on NSCs to promote regeneration [165].

Theaflavin (TFA), a polyphenolic compound, is a pigment contained in black tea known to exert antioxidant [166] and anti-inflammatory effects [167], among a number of other health benefits [168]. TFA’s mechanism of action revolves around the nuclear translocation of Nrf2, activating the Nrf2/ARE pathway to upregulate HO-1 [169]. Li et al. showed that NSCs treated with TFA can result in a decreased infarct volume and improved cognitive function through Nrf2′s protective effects against oxidative stress [170]. Additionally promising for stem cell viability is a reported dose-response relationship between TFA and B-cell lymphoma 2 (Bcl-2) overexpression to downregulate the mitochondrial apoptotic pathway in ischemic stroke models [171].

Curcumin is a natural polyphenol that is isolated from turmeric. This compound has low toxicity and owes both its antioxidant and anti-apoptotic properties to activation of the Nrf2 pathway by modulating the demethylation of CpG islands to promote gene transcription [172], which also leads to enhanced activation of HO-1 and NQO1 [173,174]. Curcumin has been shown to exert a neuroprotective effect in MCAO-induced rat models [175], and post-stroke injections reduced damage to hippocampal CA1 neurons [176]. In addition to oxidative stress, the far-reaching effects of curcumin in ischemia-reperfusion injury are involved in BBB disruption, platelet adhesion and aggregation, and immune functions [177]. Endogenously, curcumin regulates NSC proliferation, differentiation, and migration, with a number of potential applications to stroke and other neurological disorders [178]. In stem cell transplantation, embryonic stem cell exosomes loaded with curcumin resulted in significant cerebrovascular regeneration in mouse models [179].

With both treatment modalities alone showing promise in treating stroke, combining pharmaceuticals or natural products with stem cell therapy can efficiently target the Nrf2 pathway to promote broad effects on cell viability and proliferation.

### 4.3. Downstream Targets of the Nrf2 Pathway

In addition to the potential stroke therapies described above, there exist a number of compounds known to target key mediators related to the oxidative and inflammatory response that exist downstream of the Nrf2 pathway. The following will summarize a number of such treatment options while emphasizing the advantages of targeting Nrf2 upstream for broader therapeutic outcomes.

HO-1 is an intriguing target considering it has the greatest amount of antioxidant response elements on its promoter, when compared to other key downstream targets such as NQO1, GST, and γ-glutamylcysteine synthetase [180]. As discussed earlier, HO-1 metabolizes heme into various components including: ferrous iron (which promotes ferritin expression and thus iron sequestration, preventing iron-mediated cell injury) [181,182], biliverdin-IXa which is converted to bilirubin-IXa via biliverdin reductase (both components have antioxidant and anti-inflammatory properties) [183], and carbon monoxide (with anti-inflammatory, vasorelaxant, and anti-apoptotic properties) [184]. Animal studies have demonstrated protection against experimental stroke with various compounds activating HO-1 including some drugs previously mentioned: sulforaphane, Gingko biloba (EGb 761), curcumin, resveratrol, triterpenoids, and dimethyl fumarate [117,160,173,185,186,187,188,189]. Melatonin specifically increases expression of HO-1 downstream of Nrf2, resulting in improvement of motor skills and reduction in infarction size in experimental stroke models [190]. With regard to stem cells, melatonin has been shown to enhance neurogenesis in peri-infarct regions [191]. Other inducers of downstream HO-1 expression include oleanolic acid, hemopexin, and propofol, all of which have brought about improved outcomes in stroke models [192,193,194].

NQO1 is an important downstream enzyme involved in detoxifying reactive species. In rats who suffered stroke, intraperitoneal injection of 300 mg/kg curcumin was shown to induce expression of NQO1 [174], as well as reduce oxidative stress and improve binding of Nrf2 to ARE. Animal studies also showed increased concentrations of mRNA and protein product of NQO1 (as well as Nrf2, KEAP1, and HO-1) after treatment of tBHQ [154].

GSTs are enzymes that utilize the supply of reduced glutathione to detoxify xenobiotics. As previously noted, sulforaphane activates the Nrf2 pathway, inducing GSTs, as well as HO-1 and NQO-1 [28]. γ-glutamylcysteine synthetase is another enzyme important in the glutathione pathway, and its expression (along with glutathione reductase and glutathione synthetase) was seen to be induced in astrocytes and microglia in the setting of dimethyl fumarate administration [118].

NF-κB, a mediator demonstrating previously discussed antagonistic crosstalk with the Nrf2 pathway, is a key regulator of neuroinflammation that leads to the common complications of edema, hemorrhage, and necrosis following stroke [195,196,197]. For this reason, a significant body of research exists into post-stroke inhibition of the NF-κB pathway. Experimental models of cerebral ischemia have indicated that statins, such as simvastatin [198] and atorvastatin [199], substantially reduce NF-κB expression in brain tissue through transcriptional inhibition [200]. In exogenous stem cell therapy, simvastatin has been shown to aid bone-marrow-derived mesenchymal stem cell migration, while both statins have demonstrated activity in proliferation, viability, and differentiation of endogenous NSCs [201,202]. Naloxone has been shown to promote NF-κB inhibition through increased expression of the inhibitory protein, IκBα, and a reduction in NF-κB p65 nuclear translocation, resulting in decreased neuronal apoptosis and a dose-dependent decrease in infarction volume in animal models of cerebral ischemia [203]. An additional inhibitor of NF-κB nuclear translocation is artesunate, a drug prescribed for cerebral malaria that has recently found use in a mouse model of ischemic stroke, and ameliorated neuroinflammation by suppressing neutrophil infiltration and microglial activation [204]. In the context of stem cells, artesunate has been utilized to enhance NSC proliferation in the subventricular zone and peri-infarct cortex [205,206]. Aspirin, a common prophylactic in stroke prevention, may also have a role in treatment, having been shown to downregulate NF-κB-mediated endoplasmic reticulum stress in cerebrovascular endothelial cells following cerebral infarction in a mouse model [207]. Aspirin treatment alongside human umbilical cord matrix-derived stem cells improved learning and memory via the Morris water maze test [208]. Isosteviol sodium (STVNA), obtained from stevioside (a natural sweetener), exerts neuroprotective effects by interfering with the NF-κB signaling pathway [209]. In models of ischemia, STVNA has been implicated in inhibition of astrogliosis [210], modulation of microglia/macrophage polarization [211], and preservation of volume control in endothelial cells [212]. Flavonoids constitute a class of natural products with a significant role in the inhibition of microglial polarization and management of oxidative stress, with a number of compounds acting upon the NF-κB signaling pathway [213,214,215,216,217,218,219,220]. A number of the flavonoids have been implicated in the activity of stem cells, including hesperetin in NSC proliferation [221], icariin in neurogenesis and angiogenesis through release of BDNF and VEGF [222], and quercetin in enhancing stem cell viability and proliferation while reducing apoptosis [223].

While targeting of NF-κB and other downstream molecules mentioned above will certainly hold a central role in the future of stroke treatment, there exists limitations in this approach that are not yet fully appreciated. For instance, corticosteroids are commonly prescribed for controlling inflammation and have long been known as potent inhibitors of NF-κB through induction of its inhibitory protein, IκBα [224,225]. Despite this, a systematic review first published in 1997 and updated in 2011 consistently found the corticosteroids dexamethasone and betamethasone to have no effect on death or neurological/functional outcomes in acute ischemic stroke [226]. Treatment targeting Nrf2 upstream, as opposed to specific downstream targets, offers durability to treatment by enhancing a number of antioxidant, anti-inflammatory, and neuroprotective pathways should specific downstream approaches fail to achieve desired outcomes.

A comprehensive list of compounds and associated targets upstream or downstream within the Nrf2 pathway, along with the recognized effects in stroke treatment and stem cell therapy, is outlined in Table 1. Studies were completed in vivo (typically middle cerebral artery occlusion/reperfusion technique) or in vitro (typically oxygen-glucose deprivation/reoxygenation technique) in rodent models.

## 5. Conclusions and Future Directions

This review consolidates a plethora of information regarding oxidative damage and inflammation, both centrally and peripherally, following IS and HS. Furthermore, an elaborate discussion on the power of stem cells encourages ongoing studies to determine optimal safety and efficient uses of cell-based treatments for stroke. Considering both focuses, we argue that enhancing the antioxidant properties of the Nrf2 pathway concurrently with stem cell administration is a novel and imperative future research focus. Currently, the very few treatment options for IS and HS offer little reprieve from permanent functional impairments. Thus, greater initiatives are necessary to amplify treatment options for stroke. Furthermore, this review has revealed a number of potential directions for future research, including the oxidative processes that occur following cerebrovascular accident to enhance targeted therapies for improved stem cell viability and the role of Nrf2 in peripheral inflammation. Ultimately, we propose optimizing antioxidant and cell-based therapies by targeting the most upstream regulator of the antioxidant Nrf2 pathway, Nrf2 itself, to reduce secondary cell death, peripheral inflammation, and improve stem cell survival and effect.

## Figures and Tables

**Figure 1 antioxidants-11-01447-f001:**
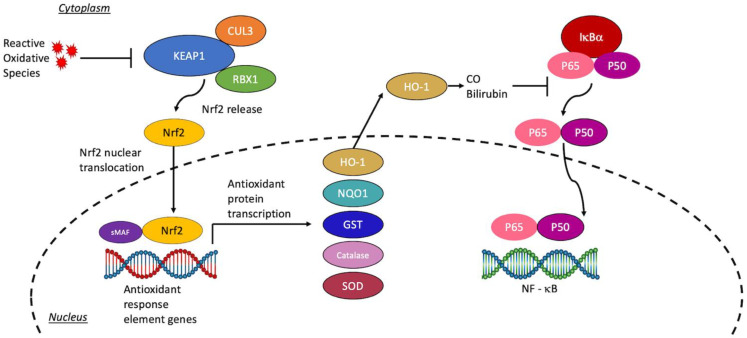
The Nrf2 pathway and downstream effectors. Note the inhibition of the KEAP1 complex by ROS to promote the release of Nrf2 for subsequent nuclear translocation and transcription of antioxidant proteins. Downstream, HO-1 inhibits NF-κB signaling to downregulate the inflammatory response.

**Figure 2 antioxidants-11-01447-f002:**
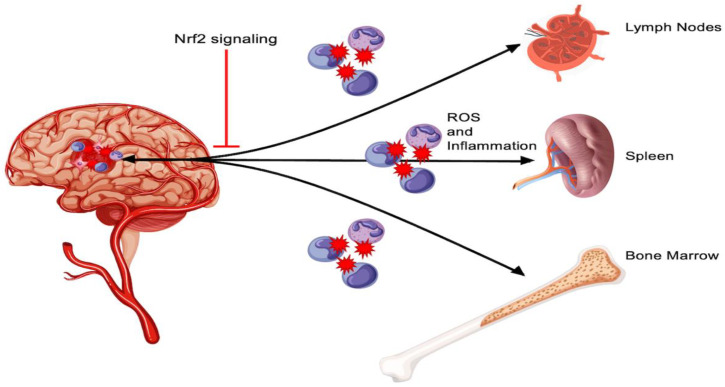
Peripheral contributions to oxidative and inflammatory responses after stroke. This figure illustrates the bidirectional inflammatory and oxidative signaling between the damaged central nervous system tissue and peripheral organs, such as the spleen, lymph nodes, and bone marrow. The Nrf2 pathway can mitigate this exacerbated response via its antioxidative protein products.

**Figure 3 antioxidants-11-01447-f003:**
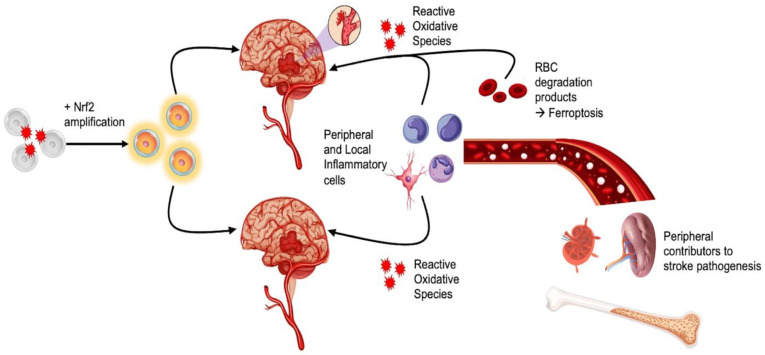
Crosstalk between Nrf2 and stem cells for targeting stroke-induced inflammation. This figure exemplifies local (neutrophils, monocytes, lymphocytes, and microglia) and peripheral (spleen, lymph nodes, and bone marrow) inflammatory and oxidative stressors due to ischemic and hemorrhagic stroke. The use of Nrf2 pathway amplification concurrent with stem cell therapy can enhance the efficacy of stem cell treatments to reduce secondary cell death following stroke.

**Table 1 antioxidants-11-01447-t001:** Nrf2-based compounds and downstream molecules that interact with stem cell therapy for stroke.

Compound	Target(s)	Effect on Stroke Treatment	Citation	Effect on Stem Cells in Setting of Stroke	Citation
Dimethyl Fumarate	Nrf2 Pathway via KEAP1	Reduced infarct volume Inhibited leukocyte infiltration Supported subcellular localization of tight junctions Decreased neurological deficits Reduced edema volume	[120]	Enhanced NSC survival Increased motor neuron survival Reduced ROS production Decreased apoptosis	[103]
Berberine	Nrf2 Pathway via PPARδ	Reduced inflammation by upregulation of inhibitory cytokines Reduced infarct volume Reduced edema volume	[227]	Promoted NSC proliferation Promoted ROS scavenging	[121]
Carbon Monoxide	Nrf2 and NF-κB Pathways	Reduced infarct volume Reduced cerebral edema Improved neurological function	[228]	Modulated NSC tolerance to iron overload Increased NSC proliferation	[126]
Sulforaphane	Nrf2 Pathway Upstream	Reduced BBB disruption Reduced lesion progression Decreased neurological deficits	[131]	Promoted NSC proliferation Promoted NSC differentiation	[132]
Doxycycline	Nrf2 Pathway Upstream	Protected BBB via inhibition of MMP-2, MMP-9, and PKCδ Blocked leukocyte adhesion	[136,137,138]	Inhibited microglial activation Promoted superoxide scavenging Reduced NSC apoptosis	[134,135]
Minocycline	Nrf2 Pathway Upstream	Preserved BBB via MMP-9 inhibition Improved neurological/functional outcomes	[143]	Increased NSC viability Increased NSC proliferation Promoted NSC release of neuroprotective factors Restored neurogenesis	[112,147]
Tert-butylhydroquinone	Nrf2 Pathway Upstream	Reduced cortical damage Reduced sensorimotor deficits	[33]	Inhibited microglial activation Decreased release of proinflammatory cytokines Increased angiogenesis	[153,155]
Resveratrol	Nrf2 Pathway Upstream	Reduced infarct volume Improved neurobehavioral scores	[159]	Increased proliferation and mobilization of endothelial cell progenitors Enhanced NSC survival and proliferation Reduced NSC apoptosis	[157,158,161]
Ginseng	Nrf2 and NF-κB Pathways	Reduced infarct volume Reduced edema volume Improved neurological outcomes	[229]	Promoted proliferation of endothelial precursor cells and NSCs Enhanced neurogenesis, angiogenesis, and synaptic plasticity Induced NSC differentiation	[164]
Theaflavin	Nrf2 Pathway Upstream (via nuclear translocation)	Reduced infarct volume and neuronal injury Improved memory impairment and learning ability	[170]	Increased Bcl-2 overexpression Inhibited mitochondrial apoptotic pathway	[171]
Curcumin	Nrf2 Pathway Upstream (via gene transcription)	Decreased neuronal cell death Decreased lipid peroxidation Protected hippocampal CA1 neurons Prevented BBB disruption	[172,175,176]	Increased NSC proliferation, differentiation, and migration Enhanced viability of embryonic stem cell exosomes	[178,179]
Gingko biloba	HO-1 Downstream	Improved infarction volume	[187]	Improved infarction volume and motor skills Enhanced proliferation of NSCs	[230]
2-cyano-3,12 dioxooleana-1,9 dien-28-oyl imidazoline	HO-1 Downstream	Upregulated HO-1 Increased neuronal survival Improved neurological dysfunction and infarct volume	[188]	-	-
Melatonin	HO-1 Downstream	Increased expression of HO-1 Improved infarct size and motor skills	[190]	Enhanced endogenous neurogenesis and cell proliferation in peri-infarct regions	[191]
Oleanolic acid	HO-1 Downstream	Attenuated cytotoxicity and overproduction of intracellular ROS via suppression of GSK-3β activation and enhancement of HO-1 expression Improved area of cerebral infarction and neurological scores	[192]	-	-
Hemopexin	HO-1 Downstream	Induced expression of HO-1 Promoted migration and differentiation of endothelial progenitor cells Facilitated angiogenesis	[193]	-	-
Propofol	HO-1 Downstream	Improved neurological deficits and infarct volume Attenuated neuron apoptosis Increased HO-1 protein expression in ischemic penumbra	[194]	-	-
Simvastatin	NF-κB Pathway	Abolished NF-κB activation	[198]	Increased bone-marrow-derived mesenchymal stem cell relocation, endogenous neurogenesis, arteriogenesis, astrocyte activation Decreased neuronal damage	[201]
Atorvastatin	NF-κB Pathway	Decreased expression of TLR4 and NF-κB Improved neurological deficit scores	[199]	Restored survival, proliferation, migration, and differentiation of NSCs	[202]
Naloxone	NF-κB Pathway	Decreased brain edema, infarction volume, and morphological injury Improved motor behavioral function Inhibited nuclear translocation of NF-κB p65 Decreased concentrations of nuclear NF-κB p65 in the ischemic penumbra Increased IκBα Attenuated phosphorylated NIK and IKKα levels in the ischemic penumbra Increased Bcl-2 and decreased Bax Stabilized mitochondrial transmembrane potential Inhibited cytochrome C release and activation of caspase 3 and caspase 9	[203]	-	-
Artesunate	NF-κB Pathway	Improved neurological deficit scores and infarct volumes Reduced neutrophil infiltration and microglia activation Downregulated TNF-α and IL-1β expression Inhibited nuclear translocation of NF-κB	[204]	Promoted proliferation of NSCs in ipsilateral subventricular zone and peri-infarct cortex	[205,206]
Aspirin	NF-κB Pathway	Suppressed TLR4 and NF-κB expression in cerebrovascular endothelial cells Improved infarct area	[207]	Improved learning and memory with human umbilical cord matrix-derived stem cells	[208]
Isosteviol sodium	NF-κB Pathway	Improved infarct volume and neurological scores Increased number of restored neurons and decreased astrocytes Downregulated mRNA expression of inhibitor of nuclear factor kappa-B kinase-α, inhibitor of nuclear factor kappa-B kinase-β, NF-κB, inhibitor of NF-κB-α, tumor necrosis factor-α, interleukin-1 beta, Bcl-2-associated X protein, and caspase 3 Upregulated mRNA of Bcl-2	[209]	-	-
Hesperetin	NF-κB Pathway	Improved neurological deficit Regulated polarization of microglia	[213]	Induced proliferation of NSCs	[221]
Baicalein	NF-κB Pathway	Improved infarct volume and sensorimotor function Decreased proinflammatory markers, release of proinflammatory cytokines, and nitric oxide Increased anti-inflammatory markers CD206 and Arg-1 Reduced TLR4, phosphorylation of IKBα and p65, and nuclear translocation of NF-κB p65 Inhibited phosphorylation of signal transducer and activator of transcription 1 (STAT1)	[214]	-	-
Icariin	NF-κB Pathway	Reduced cerebral infarct volume, neurological deficit, cerebral cell death of rats Downregulated expression of TNF-α, IL-6, C-caspase 3, and Bax Upregulated expression of Bcl-2 Downregulated activation of PPARs/Nrf2/NF-κB and JAK2/STAT3 pathways	[216]	Increased expression of BDNF and VEGF in the hippocampus and frontal cortex Promoted angiogenesis and neurogenesis Improved brain infarction volumes, motor and somatosensory deficits, and neurobehavioral outcomes	[222]
Genistein-3′-sodium sulfonate	NF-κB Pathway	Improved brain infarct volume and neurological function Reduced microglia M1 polarization and IL-1β levels Inhibited activation of NF-κB signaling in the ischemic penumbra	[218]	-	-
Quercetin	NF-κB Pathway	Improved cerebral infarct volumes Improved cognitive and motor deficits	[220]	Improved neurological functional recovery Reduced proinflammatory cytokines IL- 1β and IL-6 Increased anti-inflammatory cytokines IL-4, IL-10, and TGF-β1 Inhibited cell apoptosis Improved survival rate of human umbilical mesenchymal stromal cells	[223]
Anfibatide	NF-κB Pathway	Improved neurological deficit, neurobehavioral impairment, and infarct volume Increased cell viability Decreased LDH release Inhibited expression of p-IκBα, p-p65, NLRP3, ASC, cleaved caspase 1, Bax, and cleaved caspase3 Promoted expression of Bcl-2 Decreased TUNEL-positive cell number and concentration of IL-β and IL-18	[231]	-	-
Cyclo-(Phe-Tyr)	NF-κB Pathway	Decreased size of cerebral infarct Improved neurological scores Blocked inflammatory and oxidative factor release	[232]	-	-
Maraviroc	NF-κB Pathway	Improved neurological deficits and infarct volumes Decreased levels of apoptosis and inflammation Increased viability of primary microglia Decreased secretion of and expression of IL-1β, IL-6, and TNF-α in microglia Inhibited activity of NF-κB pathway and JNK pathway	[233]	-	-
Donepezil	NF-κB Pathway	Increased cell viability of human brain microvascular endothelial cells Promoted cell migration and angiogenesis Decreased cell permeability Increased expression of tight junction proteins Regulated expression of SIRT1/FOXO3a/NF-κB	[234]	Enhanced post-stroke neurogenic effects that naturally occur in the subventricular zone such as: ○ Upregulated metabotropic acetylcholine receptors, phosphorylated protein kinase C, and p-38 ○ Increased number of BrdU/doublecortin-positive cells, protein count of phosphorylated-neural cell adhesion molecules, and mammalian achaete scute homolog-1 Induced proliferation of NSCs and neuroblasts in subventricular zone	[235,236]
Dexmedetomidine	NF-κB Pathway	Reduced infarction area Increased miR-214 expression	[237]	-	-
Aloe-emodin	NF-κB Pathway	Improved infarct size and behavioral score Decreased expression of TNF-α, MDA, LDH, caspase 3, and NF-κB Increased expression of SOD, Bcl-2/Bax, PI3K, AKT, and mTOR	[238]	-	-
9-Methylfascaplysin	NF-κB Pathway	Improved motor impairments and infarct size Reduced activation of microglia/macrophage in ischemic penumbra Reduced expression of proinflammatory factors Inhibited oxidative stress and activation of NF-κB and NLRP3 inflammasome	[239]	-	-
Uric acid	NF-κB Pathway	Attenuated severity of cerebral infarction and activation of microglia in cerebral cortex Reduced release of proinflammatory cytokines TNF-α, IL1β, and IL6 Improved cell viability Decreased LDH release	[240]	-	-
Clinacanthus nutans	NF-κB Pathway	Inhibited IL-1β transcription Attenuated IκBα degradation Decreased production of IL-6 and TNFα	[241]	-	-
Pterostilbene	NF-κB Pathway	Improved neurological scores, edema, and infarct volume Increased number of mature neurons Decreased microglia activation Reduced iNOS and IL-1β mRNA expression Promoted IκBα expression Inhibited expression of inflammatory cytokines Suppressed NADPH activity Decreased ROS production	[242]	-	-
Salvianolic Acid B and Puerarin	NF-κB Pathway	Reduced ROS levels Inhibited apoptosis Improved mitochondrial membrane potential Improved neurological deficit scores and infarct area Inhibited expression of proinflammatory cytokines (TNF-α, IL-1β, IL-6)	[243]	Salvianolic Acid B: Induced proliferation of NSCs Improved cognitive impairment	[244]
Steppogenin	NF-κB Pathway	Inhibited nuclear translocation of NF-κB Suppressed JNK and p38 MAPK signaling	[245]	-	-
Triptolide	NF-κB Pathway	Attenuated brain infarction volume, water content, neurological deficits, and neuronal cell death rate Downregulated iNOS, COX-2, and GFAP Increased expression of Bcl-2 Suppressed Bax and caspase 3	[246]	-	-
Sitagliptin	NF-κB Pathway	Suppressed IL-6 and TNF-α Increased anti-inflammatory IL-10 Reduced neutrophil infiltration, lipid peroxides, and nitric oxide associated with replenished reduced glutathione Decreased glutamate Decreased cytochrome C and caspase 3	[247]	-	-
Fluoxetine	NF-κB Pathway	Decreased TNF-α, IL-1β, IL-6, and NF-κB subunits p65 and p50 Increased IκBα	[248]	Increased NSC differentiation Upregulated neurogenin1 expression Downregulated ERK2 phosphorylation	[249]

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
