# Peer review of "The Role of Concomitant Nrf2 Targeting and Stem Cell Therapy in Cerebrovascular Disease"

_antioxidants, 2022, doi:10.3390/antiox11081447_

Round 1
Reviewer 1 Report
This is a very interesting and well-written review that would call a great deal of interest from both research scientists and clinicians in the field of Cerebrovascular Disease. In my view, this insightful review would be informative and helpful in discovering effective therapies in treating cerebral ischemia and hemorrhagic stroke.
My concerns are as follows:
The authors stated, on the page 4 under the section of “4.1. Mechanistic interrogation of peripheral inflammation”, “Of note, the spleen has been shown to decrease in size following stroke in rat subjected to MCAO [86,89,90], and rats that underwent splenectomy prior to the MCAO showed an 82.3% decrease in infarct volume compared to those with an intact spleen [86]”; on the page 5 under the section of “4.2. Stem cells and peripheral inflammation”, “recent research has shown that there is potential for stem cell therapy to modulate the peripheral immune response. Infusion of human umbilical cord blood cells were shown to restore spleen size and function in rats with MCAO [103], and later studies showed that multipotent adult progenitor cells have been shown to suppress groups of genes that upregulate inflammatory response from within the spleen [104].” Logically and scientifically, it is necessary to include mechanistic explanations about the three observational reports: i) how did the stroke reduce the rat spleen size? ii) how did splenectomy result in the decreased infarct volume? and iii) was restoring spleen size and function beneficially for rats with MCAO? Moreover, to include a diagram/schematic about “the intricacies of the brain’s post-stroke communication with organs in the body should enhance readers’ understanding of the nature of the stroke as a multi-organ disease, thereby making the review more accessible and interesting.
The authors stated, on the page 6 under the section of “5.1. Current stem cell therapies for stroke”, “It is hypothesized that these cell-based 238 therapies work via a by-stander effect [107]”. To reduce potential confusion from readers, a brief introduction about what ‘a by-stander effect’ is would be necessary.
Because the focus of this review is about “the role of concomitant Nrf2 Targeting and stem cell therapy in cerebrovascular disease”, logically and scientifically, it is necessary to include more information as to how Nrf2-based antioxidant treatment modality can be combined with (or integrated into) stem cell-based therapy as proposed.
Author Response
Dear Editors:
Thank you so much for your constructive comments and allowing for the opportunity to improve our work. We have amended our manuscript to reflect the changes and concerns of the reviewers. Below is our point-by-point response to the review, with specific critiques and concerns directly quoted in italics, followed by our response.
“How did the stroke reduce the rat spleen size?”
Response: Thank you for your suggestion. Splenic reduction after stroke may approximate the disease-induced systemic inflammation. The decrease in spleen volume is largely due to release of lymphocytes, monocytes, and neutrophils from the spleen (J Neuroimmune Pharmacol. 2012 Dec;7(4):1017-24; Transl Stroke Res. 2016 Feb;7(1):70-8). This happens as a result of an increase in catecholamines as seen in stroke models, such as MCAO (Exp Neurol. 2009 Jul;218(1):47-55). We have integrated these details into the manuscript as requested.
“How did splenectomy result in the decreased infarct volume?”
Response: Splenocytes contribute to the exacerbation of inflammatory processes following a stroke (J Neurosci Res 2008, 86, 2227-2234). Accordingly, their removal with splenectomy may decrease infarct volume post-splenectomy (J Neuroinflammation. 2018 Jul 3;15(1):195; J Neurosci Res 2008, 86, 2227-2234). We have incorporated these spleen-brain pathological events into the manuscript as requested.
“Was restoring spleen size and function beneficially for rats with MCAO?”
Response: Targeting the spleen to normalize its size and function reduce the MCAO-induced histological and behavioral deficits. For example, transplantation of stem cells restore spleen size and function which reduced ischemic brain damage and improved behavioral performance in treated MCAO rats compared to untreated MCAO rats (Exp Neurol 2006, 199, 191-200; Stroke. 2004 Oct;35(10):2390-5). This shows that therapeutically targeting the spleen indeed benefits stroke. We have added these supporting studies into the manuscript as requested.
“To include a diagram/schematic about ‘the intricacies of the brain’s post-stroke communication with organs in the body’ should enhance readers’ understanding of the nature of the stroke as a multi-organ disease, thereby making the review more accessible and interesting.”
Response: Thank you for your suggestion. We have created a diagram to represent the interactions between different organs after a stroke to help readers better understand its complex nature. This has been added to the manuscript.
“The authors stated, on the page 6 under the section of ‘5.1. Current stem cell therapies for stroke’, ‘It is hypothesized that these cell-based 238 therapies work via a by-stander effect [107]’. To reduce potential confusion from readers, a brief introduction about what ‘a by-stander effect’ is would be necessary.”
Response: Thank you for your suggestion. In this context, as opposed to the transplanted cells replacing the dead or dying ischemic cells, the by-stander effect refers to the grafted stem cells releasing protective factors against neuroinflammation and oxidative stress (Stroke. 2004 Sep;35(10):2385-2389; Neurobiol Dis. 2010 Oct;40(1):265-276; Front Cell Neurosci. 2021 Apr;15:654290) which promotes the mentioned neurogenesis, angiogenesis, oligodendrogenesis, vasculogenesis, and synaptogenesis. This has been added to the manuscript.
“Because the focus of this review is about “the role of concomitant Nrf2 Targeting and stem cell therapy in cerebrovascular disease”, logically and scientifically, it is necessary to include more information as to how Nrf2-based antioxidant treatment modality can be combined with (or integrated into) stem cell-based therapy as proposed.”
Response: Thank you for your suggestion. We completely agree with the Reviewer that highlighting how Nrf2-based antioxidant treatment modality can be combined with stem cell-based therapy will capture the Nrf2-stem cell interaction. We refer the Reviewer to section 4.2 which describes either co-administration of Nrf2-associated drugs with exogenous stem cells, or using stand-alone Nrf2 drugs through activation of endogenous stem cells in promoting stroke recovery.
Reviewer 2 Report
In this review authors screened the current literature available on stroke-induced inflammatory and oxidative damage as a potential limit for the therapeutic use of stem cells. Moreover, authors focused on Nrf2 as endogenous antioxidant and anti-inflammatory factor highlighting nrf2 as an attractive target for novel therapeutics to enhance stem cell therapy and promote neurogenesis
Although the manuscript is clear and generally well written, the presence of many headings and subheadings makes difficult the reading and comprehension of the text. I suggest to reduce the presence of subheadings to make the text easier to read. In particular:
- Introduction: in this section authors must introduce the NRF2/KEAP1 pathway since is not even mentioned. Moreover, it deserves to be highlight that NRF2 increase is not always a cure-all since this is one of the main cause of chemoresistance onset in cancer treatment (as recently reviewed PMID: 35453348, 34070502)
- paragraphs 2.2 and 3 : these paragraphs could be simplified and moved in the introduction
- paragraph 5.1 and 5.2 may be simplified and fuse together
- paragraph 5.4: Is it necessary? the downstream targets of the Nrf2 pathway can be mentioned when discussed. For example, lines 437-442: the role of curcumin in NQO1 modulation could be introduced in lines 399-410 just saying that these beneficial effects of curcumin may be due to the modulation of NQO1.
Author Response
Dear Editors:
Thank you so much for your constructive comments and allowing for the opportunity to improve our work. We have amended our manuscript to reflect the changes and concerns of the reviewers. Below is our point-by-point response to the review, with specific critiques and concerns directly quoted in italics, followed by our response.
“Introduction: in this section authors must introduce the NRF2/KEAP1 pathway since is not even mentioned. Moreover, it deserves to be highlight that NRF2 increase is not always a cure-all since this is one of the main cause of chemoresistance onset in cancer treatment (as recently reviewed PMID: 35453348, 34070502)”
Response: We agree with your suggestion and have added an introduction to the NRF2/KEAP1 pathway in this section and included details regarding the potential limitations of NRF2 in disease outcomes.
“Paragraphs 2.2 and 3: these paragraphs could be simplified and moved in the introduction”
Response: We also concur with your suggestion. We have simplified and reorganized these sections as part of the introduction.
“Paragraph 5.1 and 5.2 may be simplified and fuse together”
Response: We follow your suggestion and have simplified and reorganized these sections into a single subheading titled “Stem cell therapies and Nrf2.”
“Paragraph 5.4: Is it necessary? the downstream targets of the Nrf2 pathway can be mentioned when discussed. For example, lines 437-442: the role of curcumin in NQO1 modulation could be introduced in lines 399-410 just saying that these beneficial effects of curcumin may be due to the modulation of NQO1.”
Response: Thank you for your suggestion. We seriously considered the Reviewer’s suggestion. Our approach is to designate sections 4.2 and 4.3 (previously 5.3 and 5.4) as distinct sections dedicated to upstream and downstream Nrf2 targeting pathways, respectively. We maintain these two sections, but also accommodated the Reviewer’s suggestion to discuss certain drugs/compounds, such as curcumin, to appear in both 4.2 and 4.3 when they affect both upstream and downstream Nrf2 pathways.
Round 2
Reviewer 2 Report
The manuscript has been significantly improved and can be accepted in the present form.